Benchmarking long-read genome sequence alignment tools for human genomics applications

LoTempio Jonathan 1 2
Delot Emmanuele 3 4
Vilain Eric evilain@hs.uci.edu 1 2
1 Institute for Clinical and Translational Science, University of California , Irvine , CA , United States of America
2 International Research Laboratory (IRL2006) “Epigenetics, Data, Politics (EpiDaPo)”, Centre National de la Recherche Scientifique , Washington , DC , United States of America
3 Center for Genetic Medicine Research, Children’s National Hospital , Washington , DC , United States of America
4 Department of Genomics and Precision Medicine, George Washington University , Washington , DC , United States of America
Zhang Xin
Electronic publication date: 2023 Dec 18
Publication date: 2023
Volume: 11
Electronic Location ID: e16515
Received 2023 Jul 25; Accepted 2023 Nov 2
Copyright: ©2023 LoTempio et al.
Copyright year: 2023
Copyright holder: LoTempio et al.
License: This is an open access article distributed under the terms of the Creative Commons Attribution License, which permits unrestricted use, distribution, reproduction and adaptation in any medium and for any purpose provided that it is properly attributed. For attribution, the original author(s), title, publication source (PeerJ) and either DOI or URL of the article must be cited.
License URL: https://creativecommons.org/licenses/by/4.0/

Keywords: Long-read sequence, Genome alignment, Benchmark, Genomic medicine, Short-read sequence

Funding: Molecular Genetics at Children’s National Hospital The National Human Genome Research Institute U01HG011745 The University of California, Irvine’s Institute for Clinical and Translational Science, and the National Center for Advancing Translational Sciences UL1TR001414 Eric Vilain was supported by the A. James Clark Distinguished Professorship in Molecular Genetics at Children’s National Hospital. Eric Vilain, Jonathan LoTempio, Emmanuele Delot are part of the GREGoR consortium, supported by NIH Award Number U01HG011745 from the National Human Genome Research Institute. The University of California, Irvine’s Institute for Clinical and Translational Science, and the National Center for Advancing Translational Sciences (UL1TR001414) supported the publication fee for this article. The funders had no role in study design, data collection and analysis, decision to publish, or preparation of the manuscript.

==============================
Background

The utility of long-read genome sequencing platforms has been shown in many fields including whole genome assembly, metagenomics, and amplicon sequencing. Less clear is the applicability of long reads to reference-guided human genomics, which is the foundation of genomic medicine. Here, we benchmark available platform-agnostic alignment tools on datasets from nanopore and single-molecule real-time platforms to understand their suitability in producing a genome representation.

Results

For this study, we leveraged publicly-available data from sample NA12878 generated on Oxford Nanopore and sample NA24385 on Pacific Biosciences platforms. We employed state of the art sequence alignment tools including GraphMap2, long-read aligner (LRA), Minimap2, CoNvex Gap-cost alignMents for Long Reads (NGMLR), and Winnowmap2. Minimap2 and Winnowmap2 were computationally lightweight enough for use at scale, while GraphMap2 was not. NGMLR took a long time and required many resources, but produced alignments each time. LRA was fast, but only worked on Pacific Biosciences data. Each tool widely disagreed on which reads to leave unaligned, affecting the end genome coverage and the number of discoverable breakpoints. No alignment tool independently resolved all large structural variants (1,001–100,000 base pairs) present in the Database of Genome Variants (DGV) for sample NA12878 or the truthset for NA24385.

Conclusions

These results suggest a combined approach is needed for LRS alignments for human genomics. Specifically, leveraging alignments from three tools will be more effective in generating a complete picture of genomic variability. It should be best practice to use an analysis pipeline that generates alignments with both Minimap2 and Winnowmap2 as they are lightweight and yield different views of the genome. Depending on the question at hand, the data available, and the time constraints, NGMLR and LRA are good options for a third tool. If computational resources and time are not a factor for a given case or experiment, NGMLR will provide another view, and another chance to resolve a case. LRA, while fast, did not work on the nanopore data for our cluster, but PacBio results were promising in that those computations completed faster than Minimap2. Due to its significant burden on computational resources and slow run time, Graphmap2 is not an ideal tool for exploration of a whole human genome generated on a long-read sequencing platform.

Introduction

The diverse ecosystem of DNA preparation, sequencing, and mapping technologies is capable of generating computational representations of genome biology through different chemistries and processes on different sequencing and mapping platforms (Sanger, Nicklen & Coulson, 1977; Fuller et al., 2009; Branton et al., 2008; Ardui et al., 2018; Levy-Sakin et al., 2019). This has resulted in many different approaches for making use of the data of different sources, with algorithms and the tools built upon them used across platforms to differing success. Here we briefly outline the differences between these platforms and analysis strategies so that we may consider an important gap in applying the newest technologies to the problem of human genomics.

Sequencing-by-synthesis platforms produce highly accurate sequence data in the form of short-reads (<300 base pairs, bp) (Modi et al., 2021) from high molecular weight DNA inputs. In contrast, single-molecule real-time platforms can produce highly accurate reads through circular consensus sequencing (CCS) on molecules >10 kilobase pairs (kbp) (Wenger et al., 2019) while nanopore-based sequencing (Jain et al., 2018a; Jain et al., 2018b) and nanochannel-based mapping platforms (Formenti et al., 2019) can sequence or visualize megabase-length DNA molecules (Pollard et al., 2018). Each of these platforms has different utilities and available tools, which contributes to the diversity of projects enabled by these technologies (Ho, Urban & Mills, 2020; Mahmoud et al., 2019).

Optical genome mapping (developed by Bionano Genomics, San Diego, CA, USA) excels at detecting structural variants (SV), such as balanced translocations and deletions/insertions in the 1 kb to 1 Mb range, and its clinical utility was demonstrated in Duchenne (DMD) or facioscapulohumeral (FSHD) muscular dystrophies (Barseghyan et al., 2017; Sharim et al., 2019), and cancer (Neveling et al., 2021; Talsania et al., 2022; Bornhorst et al., 2023). But long-read sequence (LRS), developed by Pacific Biosciences (Menlo Park, CA, USA) and Oxford Nanopore Technologies (Oxford, UK) among others, has been shown to be the most appropriate technology to detect smaller variants in the 50 bp-1 kb range (Chaisson et al., 2019). This is especially true in repetitive regions of the genome and should therefore bring new diagnosis potential for diseases of trinucleotide repeat expansion and genome instability such as Huntington’s disease and myotonic dystrophies (Kantartzis et al., 2012; Santoro et al., 2017; Liu, Gao & Wang, 2017; Mitsuhashi et al., 2019; Mantere, Kersten & Hoischen, 2019), as well as for discovery of SVs affecting regulatory regions of known genes. Fulfilling the promise of LRS for medical genetics requires understanding the tools that power the technology and their respective strengths.

LRS platforms have been further shown to be capable of generating single, phaseable reads spanning repetitive or complex genomic regions that remain unresolved in all (Wenger et al., 2019; Jain et al., 2018a; Jain et al., 2018b; Huddleston et al., 2017; Ebbert et al., 2019) but one current human genome assembly (Nurk et al., 2022). This has implications for resolution of highly similar pseudogenes and large SV, even in low-complexity genomic regions, or from otherwise healthy diploid samples. Read lengths in the tens of kbp have further allowed for end-to-end viral genome sequencing (Walker et al., 2020), while read lengths in the millions of bp have the potential to span whole mammalian chromosomes (Jain et al., 2018a; Jain et al., 2018b; Miga et al., 2020). This potential for more contiguous de novo human assembly has led to studies to specifically improve and benchmark basecalling and polishing tools (Wick, Judd & Holt, 2019), as well as assembly tools (Koren et al., 2018; Cheng et al., 2021; Kolmogorov et al., 2019) that can be applied to human genomics.

For reference genome-guided experiments, LRS has proven useful for amplicon sequencing in cancer detection (Norris et al., 2016) and for metagenomics (Deshpande et al., 2019; Cuscó et al., 2019; Sanderson et al., 2018), but the field is still in early days for assessment of variation in high-coverage human whole genome sequence. There has been a demonstration of the added value of LRS in resolving SV has been demonstrated by efforts aggregating callsets across platforms and technologies to deeply characterize a genome (Chaisson et al., 2019). These experiments are concerned mainly with understanding the full scope of the architecture of a genome, rather than contrasting differences between results of one alignment tool or another. Each published tool showcases its own utility and strengths in their initial publications, especially in terms of the mapping quality of reads to the reference genome and precision and sensitivity to preserve known variants in synthetic or downsampled genomic data. However, there have been no studies that specifically benchmark LRS alignment platforms and tools for reference-guided experiments.

To address this gap, we benchmarked the most recent LRS alignment tools with the datasets generated from the Joint Initiative for Metrology in Biology’s Genome in a Bottle Initiative (GIAB), specifically samples, NA12878 sequenced with nanopore technology and NA24385 sequenced with CCS (branded HiFi) technology. These were generated with Oxford 9.4 pore chemistry and with Guppy5 basecalling performed by the Whole Genome Sequencing Consortium (Jain et al., 2018a; Jain et al., 2018b) and with Pacific Biosciences RSII SMRT CCS technology by PacBio and the National Institute of Standards and Technology (Zook et al., 2019). All alignments were performed on GRCh38, rather than the more complete CHM13 or more diverse pangenome graphs because there are not yet high-quality GIAB-like reference materials or variant truthsets that allow for this sort of benchmarking against these cutting-edge reference materials.

We compared computational performance (peak memory utilization, central processing unit (CPU) time, file size/storage requirements), genome depth and basepair coverage, and quantified the reads left unaligned in any given experiment. We have limited this study to tools that are platform agnostic and function on our cluster. Since the resolution of large SVs is a key application of this technology, and also allows comparison of the differences in genome alignments in an aggregate way, we ran the SV-calling tool Sniffles to highlight differences in breakpoint location in each binary alignment map (BAM) file (Sedlazeck et al., 2018). Taken together, these experiments present a comprehensive view of differences in the products of LRS whole-genome alignment pipelines.

Materials & Methods

Portions of this text were previously published as part of a preprint (https://www.biorxiv.org/content/10.1101/2021.07.09.451840v3.full) (LoTempio, Délot & Vilain, 2023).

Tool selection criteria

Starting with tools that were recommended by the developers of each platform, we examined the tools that were cited or benchmarked against new, platform-specific tools. We also used the website Long-Read Tools (https://long-read-tools.org/), searching their database with the filters “nanopore”, “pacbio”, and “alignment”. Since this yielded many software tools, we dove deeper and excluded any tools not designed for whole genome experiments. Tools which passed this test were then assessed for their suitability for use with nanopore and SMRT data and whether they produced a SAM/BAM file for analysis. Since tools can be regularly updated or out-versioned, we wanted to use only the most up to date software at the time of analysis.

In our examination of structural variant calling pipelines, we elected to limit our study to the variant caller sniffles due to the fact that the output files were most descriptive. Specifically, Sniffles outputs VCFs which call SVs as indels, translocations, duplications, inversions, and more classifications. This increases its utility for comparison to reference truthsets.

Data

Main experiments

Reference genome: GRCh38 was accessed on March 22, 2022 from NCBI (GRCh38_no_alt_analysis_set).

Sequence data

No new or original sequence data from new or existing samples were generated here. Instead, we leveraged data generated by third party consortia on well-characterized samples. These data include nanopore sequence data generated on NA12878 from the WGS Consortium, rel7 guppy 5 basecalls was accessed on March 22, 2022 (NA12878 rel7).

We also used the 15kb insert size SMRT CCS data from sample NA24385 was accessed from the human pangenomics consortium GitHub on March 22, 2022 (NA24385 15kb insert).

While no DOI exist for these files, the citations provide the most stable URLs available (PacBio, 2021).

SV truthsets

Structural variant truthsets for NA12878, updated February 25, 2020, from DGV were accessed on March 31, 2022 (NA12878 DGV truthset).

Structural variant truthsets from Pacific Biosciences and the GIAB consortium were accessed on May 20, 2022 (PacBio GIAB SV Truthset).

Absolute coordinate systems within a reference genome have allowed for very accurate genome mapping and analysis within that reference. However, reference sets are tied to these absolute coordinate systems and require liftover and curation. Our study is subject to the limitation that these variants were called with alignments to GRCh37, but we used the more advanced GRCh38. Accordingly, the absolute coordinates of the location of breakpoints differs across builds. By focusing our study on the size of variants within 1,000 and 10,000 bp windows, rather than their relative position within a set of reference genome coordinates, we are confident that this is a minor limitation.

Pilot experiments

Initial pilot experiments were performed on older reference builds and datasets. These can be found as Supplement S4 and may be useful to investigators who have LRS data from the previous decade.

Reference genome

GRCh38.p12 was accessed and downloaded on April 8, 2019 (GRCh38.p12).

Sequence data

Nanopore sequence data were accessed on April 8, 2019, version rel5-guppy-0.3.0-chunk10k.fastq, from AWS Open Data as made available by the Whole Genome Consortium. SMRT sequence data were accessed on May 28, 2019, version sorted_final_merged.bam from the NCBI FTP (NA12878 rel 5; NA12878 SMRT Mt Sinai).

Database of genome variants

DGV data were accessed on January 30, 2020 (Database of Genomic Variants, DGV). The full set of variants was reduced to 11,042 variants confirmed to be present in NA12878. A total of 14 of these variants were excluded as they were called on contigs present out of the main reference assembly contigs.

Hardware configuration

Computations were performed on the George Washington University High Performance Computer Center’s Pegasus Cluster on SLURM-managed default queue compute nodes with the following configuration: Dell PowerEdge R740 server with Dual 20-Core 3.70 GHz Intel Xeon Gold 6148 processors, 192 GB of 2666 MHz DDR4 ECC Register DRAM, 800 GB SSD onboard storage (used for boot and local scratch space), and Mellanox EDR InfiniBand controller to 100 GB fabric.

Whole genome alignment tool benchmarking and analysis

The following tool versions were used in the work presented in the main tables and figures.

1. Long-read aligner (LRA) v1.3.3. (Ren & Chaisson, 2021)

2. GraphMap2 v0.6.3 (Marić et al., 2019)

3. Minimap2 v2.24 (Li, 2018)

4. CoNvex Gap-cost alignMents for Long Reads (NGMLR) v0.2.7 (Sedlazeck et al., 2018)

5. Winnowmap v2.0.3 (Jain et al., 2022)

The versions of alignment tools used in the preliminary experiments of this project available in S4 included:

1. GraphMap v0.5.2 (Sović et al., 2016)

2. Minimap2 v2.16 (Li, 2018)

3. NGMLR v0.2.7 (Sedlazeck et al., 2018)

All tools were run with recommended parameters and were flagged for nanopore or SMRT data based on the requirements of that experiment. Where multithreading was available, -t, or equivalent was set to 40, the max for our cluster. LRA, Minimap2, NGMLR, and Winnowmap2 all have specific presets for SMRT and nanopore-based data while Graphmap2 uses the same parameterization for all LRS platforms, with an option for alternative defaults to handle Illumina data. Its default parameters are optimized for accuracy and through graph-based handling of reads can address complex genomic regions and users have the option of modifying parameters to decrease accuracy.

LRA uses the same minimizers as seeds for both sequencing platforms, but decreases the density of minimizers in the genome by nearly 10-fold, which are then compared to minimizers from the reference, resulting in a set of query-reference anchors. Anchors are clustered based on read accuracy, with lower accuracy nanopore reads given a minimum of three clusters per read, and higher accuracy SMRT-CCS reads given a minimum of 10 clusters per read.

Minimap2, upon which Winnowmap2 builds, uses homopolymer-compressed minimizers in the seeding step of its algorithm. This is to account for errors in SMRT movie data which can struggle with homopolymer runs. For example, GATTAACCA becomes GATACA through homopolymer minimization. The nanopore default parameterization uses ordinary minimizers as seeds because in development, they saw no increased benefit to homopolymer compression, even though the technology has similar limitations to SMRT. Winnowmap2 builds on this by down-weighting frequently occurring minimizers to minimize masking seeds in complex genomic regions including tandem repeats.

For NGMLR generally, reads are handled in a way which is SV-aware and accounts for small (10 bp) indels or split reads over larger indels. Reads which can be mapped linearly are, and the remaining reads are handled through splitting them over SV. The nanopore preset further customizes parameters to address the characteristic high error rates, including substitutions, insertions, and deletions, inherent to nanopore sequencing. It achieves this by lowering match/mismatch scores, increasing gap penalties, and emphasizing sensitivity, making it well-suited for applications like structural variant analysis. Conversely, the SMRT preset retains higher match/mismatch scores to prioritize base accuracy, suitable for PacBio data known for lower substitution errors but higher indel error rates.

Computational metrics were printed from SLURM job records with seff and sacct with flags–format JobID, JobName, Elapsed, NCPUs, TotalCPU, CPUTime, ReqMem, MaxRSS, MaxDiskRead, MaxDiskWrite, State, ExitCode. Samtools 1.15.1 was used for all alignment manipulations, alignment read depth coverage calculations, and to extract unmapped reads. Samtools view -f 4 was used to generate bamstats files and Python Venn was used to compare the readnames across unmapped read files to assess the degree of overlap of unmapped reads across these subsetted alignment files (Li et al., 2009; Danecek et al., 2021). R 3.5.2 version Eggshell Igloo with the tidyverse packages were used for preliminary experiments on depreciated data shown in S4. Genome coverage was calculated with samtools coverage. Binary conversion values were used for bytes (1,073,741,824) and kilobytes (1,048,576) to gigabytes.

Data reshaping and visualization

As data were integrated across multiple sources and formats, they needed to be reshaped for comparison and visualization. For example, the multiple separators found in VCFs (commas and semicolons) are not directly usable in python pandas. Relevant dataframes from genome alignment files were reshaped with shell scripts and Python scripts whose methodology and key intermediate files are available on our GitHub.

Comparison of breakpoints

Structural variants (SV) were called with sniffles/1.0.11 with default parameters (Sedlazeck et al., 2018). Only variants called on the main GRCh38 assembly were included. Python was used to bin and graph structural variants by SV Type. Intermediary files and scripts are available at our GitHub, archived stably here: https://zenodo.org/badge/latestdoi/429362081.

Data from Sniffles VCFs and from DGV were subsetted by comparable fields including SV Type, SV Length, and Chromosome since a common format was unavailable for direct comparison. Figures were made with seaborn and matplotlib as well as Microsoft Excel (Waskom, 2021; Hunter, 2007).

Results

Rigorously annotated variants were drawn from the Database of Genomic Variants (DGV) (MacDonald et al., 2014) for NA12878 and from a curated public repository for NA24385 (PacBio, 2021).

We present our results in four sections:

(1) the tools that were included,

(2) computational performance and benchmarking,

(3) an analysis of aligned and unaligned reads,

(4) an analysis of structural variation present in each alignment compared to a baseline.

Tools that passed the inclusion/exclusion criteria

Following a literature review of available alignment tools and search of Long-Read-Tools (Amarasinghe, Ritchie & Gouil, 2021), we established a set of inclusion criteria (see Material and Methods), which accounted for both types of LRS data (a tool must be able to handle both nanopore and SMRT reads), as well as the state of the field in terms of software updates (must not have been superseded by another tool). All relevant tools and their reason for exclusion are outlined in Table 1, with all tools annotated as platform agnostic and relevant to genome alignment included in Supplemental S1. Five alignment tools, GraphMap2, LRA, minimap2, NGMLR, and Winnowmap2 were included in this study (Marić et al., 2019; Li, 2018; Sedlazeck et al., 2018; Jain et al., 2022; Ren & Chaisson, 2021).

Table 1 Long-read genome sequence alignment tools.

The tools included in the study are highlighted in this study. Further tools from Long-read-tools.org are available in S1.

Tool	Year	PMID	DOI	Inclusion	If no, why?	
BLASR	2012	22988817	10.1186/1471-2105-13-238	No	Designed only for SMRT seq	
BWA-MEM	2011	NA	arXiv:1303.3997	No	Superseded by Minimap2	
GraphMap	2016	27079541	10.1038/ncomms11307	No	Superseded by GraphMap2	
GraphMap2	2019	NA	10.1101/720458	Yes	 	
Kart	2019	31057068	10.1142/S0219720019500082	No	Designed only for SMRT, SBS	
LAMSA	2016	27667793	10.1093/bioinformatics/btw594	No	Depreciated dependencies resulting in seg fault	
LAST	2011	21209072	10.1101/gr.113985.110	No	Not parameterized for the nuances of current platforms	
lordFAST	2018	30561550	10.1093/bioinformatics/bty544	No	Runs resulted in partial completion and a seg fault	
LRA	2021	34153026	10.1371/journal.pcbi.1009078	Yes	 	
Mashmap2	2017	30423094	10.1093/bioinformatics/bty597	No	Does not use SAM format needed for variant caller	
MECAT	2017	28945707	10.1038/nmeth.4432	No	Superseded by Mecat2	
MECAT2	2017	28945707	10.1038/nmeth.4432	No	Designed only for SMRT seq	
Meta-aligner	2017	28231760	10.1186/s12859-017-1518-y	No	Designed only for SMRT seq	
Minimap2	2018	29750242	10.1093/bioinformatics/bty191	Yes	 	
NGMLR	2018	29713083	10.1038/s41592-018-0001-7	Yes	 	
rHAT	2015	26568628	10.1093/bioinformatics/btv662	No	Designed only for SMRT seq	
S-ConLSH	2021	33573603	10.1186/s12859-020-03918-3	No	Designed only for SMRT seq	
smsMap	2020	32753028	10.1186/s12859-020-03698-w	No	Does not use SAM format needed for variant caller	
Winnowmap	2020	32657365	10.1093/bioinformatics/btaa435	No	Superseded by Winnowmap2	
Winnowmap2	2022	35365778	10.1101/2020.11.01.363887	Yes	 	
YAHA	2012	22829624	10.1093/bioinformatics/bts456	No	Not parameterized for the nuances of current platforms	

Sixteen tools were excluded because they were superseded by other tools, produced an alignment file in a non-SAM format, were designed for only one type of input data, or did not work on our cluster (Kiełbasa et al., 2011; Li, 2013; Chaisson & Tesler, 2012; Faust & Hall, 2012; Liu et al., 2016; Sović et al., 2016; Liu, Gao & Wang, 2017; Jain et al., 2018a; Jain et al., 2018b; Nashta-Ali et al., 2017; Xiao et al., 2017; Haghshenas, Sahinalp & Hach, 2019; Li, 2018; Sedlazeck et al., 2018; Marić et al., 2019; Kumar, Agarwal & Ranvijay, 2019; Wei, Zhang & Liu, 2020; Jain et al., 2020; Jain et al., 2022; Ren & Chaisson, 2021; Chakraborty, Morgenstern & Bandyopadhyay, 2021).

We excluded LAST and YAHA as they were not parameterized for the nuances of current sequencing platforms, i.e., they did not have default settings that took into account long reads and low error rates because their success would require more careful consideration of research questions, rather than exploration. smsMap produced an output that would require post-processing to put it into a SAM compatible format, and since we did not aim to create new software here, we excluded it. LAMSA, lordFAST and were run, but resulted in segmentation faults (Haghshenas, Sahinalp & Hach, 2019; Liu, Gao & Wang, 2017). LAMSA was built to call precompiled GEM3 libraries, with which our current compute architecture was incompatible, while lordFast produced an undiagnosable error log (Marco-Sola et al., 2020). The other 11 tools, outlined in Table 1, were excluded because they were only designed to handle data from one LRS platform.

Computational performance and benchmarking

Computations were run three times on a single node allocated fully on our university’s high-performance compute cluster (HPC), which includes 40 CPUs with a configuration described in the methods. All tools were run with default parameters to reflect typical use in exploratory studies. High-level computational benchmarks can be found in Table 2. Full reports on each run can be examined in Supplemental S2, while a composite table of the output of samtools coverage is available as Supplemental S3.

Table 2 Benchmarking metrics.

Each of the three runs for each tool is shown in this table. All values are taken directly from slurm’s sacct and seff features. All metrics are intuitive except for CPU efficiency, which is a measure of idle CPUs to CPU time over wall clock time.

Platform	Tool	Wall clock time
(D-H:M:S)	CPU time
(D-H:M:S)	CPU
efficiency	Memory utilized
(Gb)	
Nanopore	Graphmap2	14-00:00:04	560-00:02:40	65.25%	90.19	
14-00:00:17	560-00:11:20	63.34%	94.58	
14-00:00:15	560-00:10:00	64.23%	89.68	
Minimap2	1-11:00:08	58-08:05:20	7.51%	12.43	
1-10:48:18	58-00:12:00	7.51%	12.80	
1-11:07:00	58-12:40:00	7.51%	12.78	
NGMLR	10-00:00:03	400-00:02:00	97.70%	35.86	
10-00:00:10	400-00:06:40	99.56%	34.38	
9-14:55:56	384-21:17:20	93.43%	36.34	
Winnowmap2	15:25:58	25-17:18:40	83.82%	29.48	
15:19:14	25-12:49:20	83.61%	30.02	
15:15:24	25-10:16:00	83.06%	28.74	
SMRT-CCS	Graphmap2	5-01:33:22	202-14:14:40	98.88%	45.07	
5-02:54:53	204-20:35:20	98.81%	44.90	
5-06:49:25	211-08:56:40	98.52%	48.96	
LRA	6:53:00	11-11:20:00	33.20%	20.63	
6:53:04	11-11:22:40	33.11%	21.10	
6:32:33	10-21:42:00	33.22%	27.91	
Minimap2	11:02:04	18-09:22:40	7.51%	9.10	
11:06:03	18-12:02:00	7.51%	9.13	
10:55:09	18-04:46:00	7.52%	9.09	
NGMLR	21:50:29	36-09:39:20	99.20%	25.53	
22:24:38	37-08:25:20	99.19%	25.69	
21:58:53	36-15:15:20	99.53%	25.82	
Winnowmap2	3:39:42	6-02:28:00	88.10%	18.57	
3:43:42	6-05:08:00	88.32%	18.71	
3:40:32	6-03:01:20	87.91%	19.14	

Globally successful tools

Minimap2 was the least memory-demanding tool

Minimap2 successfully aligned nanopore data every time with unrestricted and restricted resources. Unrestricted runs used around 12.6  gigabytes (Gb) and the jobs took ∼34 wall clock hours to complete.

Minimap2 also successfully aligned SMRT data every time with unrestricted or restricted resources. The runs used around 9 Gb of memory and the jobs took ∼11 wall clock hours to complete, considerably faster than the (admittedly larger) nanopore dataset.

Memory usage and runtime were consistent across triplicate runs with unrestricted resources and did not change with restriction of resources when the tool was used with either dataset. The consistency of results, as well as the speed and relatively low computational demands of Minimap2 make it a strong candidate for inclusion in clinical analysis pipelines.

Winnowmap2 was the fastest tool

Winnowmap2 was the fastest to run on any dataset. Specifically, it aligned the entire nanopore dataset in around 15 wall clock hours, using only a little more than two times more memory than minimap2 ( ∼29 Gb). For SMRT, it was even faster at just over 3.5 wall clock hours and almost exactly twice as much memory ( ∼18 Gb). These numbers were consistent over all runs.

Additionally, the high CPU efficiency of the jobs on our cluster were positive to note: over 80% efficiency, relative to Minimap2’s ∼7% efficiency. This is a measure of the ratio of CPU time used to the wall clock time times the number of CPUs. In the case of our cluster, CPUs are fully allocated to a node and not shared once allocated, but it does point the way towards either choosing more efficient tools, or optimizing jobs.

NGMLR completed 4/6 tasks and performance demanded great resources

NGMLR successfully aligned SMRT data every time but was the second slowest. The runs used around 25 Gb and the jobs took around 22 wall clock hours to complete, nearly twice as long as the next tool, Minimap2.

NGMLR was much more inconsistent on nanopore data. It successfully aligned the data on our first range finding experiment in approximately ∼230 wall clock hours. Following this run, two jobs were set with a time of 10 days, and both resulted in timeouts. The single successful run used 36.34 Gb of memory, the most of all tools which produced an alignment.

Tools that presented a challenge

LRA did not work on nanopore-generated data

While LRA did show promise on SMRT data, aligning a genome in ∼11 CPU days or ∼6 wall clock hours, we could not run it successfully with nanopore. Each run resulted in unresolvable (undescriptive) segmentation faults that produced no partial alignment. In the case of SMRT data, it did fall in the front of the pack, aligning genomes quickly with typical memory usage relative to its peers ( ∼20–27 GB).

GraphMap2 was the most resource-intensive tool

GraphMap2 took the longest, used the most memory, and had the most failures of all of the tools we examined. It did not produce alignment files on nanopore data in the time limit of 14 days imposed by our HPC. On the SMRT dataset which it was successful at aligning, it took more than 5 times longer to produce an alignment file than the next slowest tool, NGMLR. It had the highest memory usage, in excess of 89 GB nanopore on 44 GB on SMRT.

Whole genome alignment

Aligned reads show disagreement in coverage of each chromosome

In preliminary experiments that were completed successfully, alignments of the same genome with the same tool had the same genome coverage, whether resource-restricted or not (S4, column O). For this reason, one file generated from the first run was used for subsequent analysis, shown in Table 3, a summary of the coverage depth of each chromosome alignment and the percentage of the basepairs of GRCh38 covered. Global genome coverage of ∼30x was reported for nanopore and ∼34x, 20x of which is present in the 15 kbp insert set, was reported for SMRT data sets. Breaking down the summary into chromosomes allows for more precision than referring to a genome by its coverage as a uniform metric, since there is discrepancy across the chromosomes.

Table 3 Sequencing depth and reference coverage.

Chromosome level descriptions of the average number of reads covering each basepair, as well as the percentage of basepairs of the reference covered. The highest value for a chromosome is accented with bold text, while the lowest value is accented with italic text. Winnowmap2 abbreviated as WM2.

	SMRT-CCS			Nanopore	
 	Mean depth (x-depth)		Coverage of basepairs (%)		 	Mean depth (x depth)		Coverage of basepairs (%)	
 	LRA	Minimap2	NGMLR	WM2		LRA	Minimap2	NGMLR	WM2		 	Minimap2	NGMLR	WM2		Minimap2	NGMLR	WM2	
chr1	19.0627	20.0049	17.9711	20.002		91.7893	92.3244	91.1358	92.3622		chr1	37.7867	33.3715	37.4783		92.4802	91.4793	92.4861	
chr2	20.0613	20.1343	19.9393	20.1463		98.7397	98.8041	98.707	98.6286		chr2	40.1079	36.3557	39.497		99.0601	98.8241	98.8272	
chr3	20.1303	20.5557	20.0446	20.3575		98.984	99.5552	98.7438	99.5645		chr3	37.7894	36.5499	39.9339		99.773	98.8066	99.767	
chr4	20.4284	20.7247	20.1254	21.8975		99.1528	99.0963	98.6843	99.2809		chr4	39.8297	36.1401	39.6729		99.4841	98.7817	99.5176	
chr5	20.0292	20.3439	19.7821	20.0655		98.2323	98.1787	97.6187	98.2961		chr5	37.1564	35.9384	37.17		98.5218	98.0472	98.5096	
chr6	19.9858	20.4916	19.9745	20.4963		98.795	99.0781	98.674	99.2314		chr6	37.4983	36.4317	38.3408		99.4746	98.9018	99.4822	
chr7	19.5124	19.7661	19.3123	19.7686		98.1498	99.2092	97.7858	98.9308		chr7	37.5093	35.7748	37.3072		99.6968	98.4965	99.6698	
chr8	20.0067	20.279	19.7645	20.272		98.5302	99.2526	98.1774	99.0982		chr8	37.2117	35.8164	37.4528		99.6024	98.4431	99.6027	
chr9	17.085	17.2704	16.5087	17.2736		86.2352	86.2731	85.4122	86.3347		chr9	33.7857	31.5634	33.62		87.3625	86.6206	87.4715	
chr10	19.8087	20.0826	19.4999	20.5247		98.2398	98.8789	98.1124	98.8153		chr10	37.9547	35.509	37.9365		99.3546	98.2053	99.3873	
chr11	19.5361	19.8058	19.4349	19.8105		98.0161	99.1255	98.1379	98.264		chr11	37.5146	35.9223	37.282		99.562	98.5893	99.0318	
chr12	19.6294	19.9493	19.6137	19.9618		98.3953	99.548	98.151	99.3563		chr12	37.9345	36.4544	37.6064		99.8008	98.397	99.7298	
chr13	17.5257	18.1764	16.9585	17.6178		85.2628	85.3114	84.1626	85.3793		chr13	32.6205	30.5633	32.35		85.6486	84.3648	85.6546	
chr14	16.815	16.8162	16.6225	16.9593		82.5962	82.5668	82.5949	82.5858		chr14	31.2147	30.4015	31.9599		82.5971	82.5928	82.5975	
chr15	16.0919	16.3256	15.7467	16.3303		80.7849	81.7561	80.0048	81.9674		chr15	32.9232	30.434	32.7613		82.7815	80.8381	82.8274	
chr16	18.4585	19.5281	18.1335	19.4345		89.1223	90.0939	87.9862	90.1285		chr16	36.9247	34.5339	37.2781		90.5049	88.4413	90.5183	
chr17	18.0451	18.8278	18.0519	18.7767		95.1484	97.5541	94.7803	95.9119		chr17	37.2793	35.2194	36.9045		97.3704	95.1519	98.046	
chr18	19.4328	19.7389	18.8198	19.7815		96.004	98.3045	94.3649	97.8677		chr18	36.8051	34.2153	36.5148		99.3802	94.1824	99.0731	
chr19	16.8558	16.8674	16.5895	16.8926		95.0649	95.0588	94.9148	95.0692		chr19	36.3404	33.9213	39.4614		95.1037	95.0137	95.1095	
chr20	19.34	20.0554	19.0343	20.1369		96.0126	98.1195	95.5606	98.0231		chr20	39.3238	35.3079	39.1318		99.1143	96.3336	99.1551	
chr21	18.0549	18.4984	16.8995	18.2805		80.4792	80.4144	78.9093	80.16		chr21	35.4168	30.0951	34.298		81.3939	80.763	81.2955	
chr22	13.9854	14.0173	13.3346	17.4726		72.7139	72.6378	72.5114	72.7444		chr22	32.7008	25.9485	31.1753		72.9147	72.9092	72.9018	
chrX	9.8603	9.97983	9.73401	9.98727		97.639	97.9196	96.4952	98.0195		chrX	36.1372	33.9833	35.6273		99.1734	97.0137	98.9417	
chrY	7.94848	8.21933	3.85465	8.37289		40.9697	40.9213	37.9947	40.8261		chrY	1.8305	0.501775	1.3745		3.1548	4.9006	2.37513	
chrM	1010.03	1203.88	1203.67	1202.19	 	100	100	100	100		chrM	11954.1	11413.1	11685.6	 	100	100	100	

For nanopore data, three alignment files were measured, while for SMRT data, four files could be included. Table 3 shows cover of each chromosome with high values represented in bold text, and low values represented in italic text. It becomes immediately apparent that Minimap2 and Winnowmap2 both retain the most reads (highest x coverage) and cover the most basepairs of the reference. NGMLR excludes the most reads and covering the fewest basepairs of the reference genome, which may impact downstream analysis. The final tool, LRA, which only worked on SMRT data, is intermediate with some lower values of coverage depth, and some higher values of coverage of basepairs.

Unaligned reads reveal differential exclusion of reads

The readname assigned to each read in a fastq retained in the BAM allowed us to directly compare the lists of reads that were not included in the alignment by each alignment tool with Python Venn (Fig. 1) (Grigorev, 2018).

Figure 1 Unmapped reads.

Venn diagrams show the total number of reads flagged as unmapped in the sorted BAM files. The left panel presents unmapped reads from the SMRT dataset. The right panel presents unmapped reads from the nanopore dataset.

All tools agreed to leave ∼1 million of the same nanopore reads unaligned, but differed in their overall totals of unaligned reads. NGMLR left the highest number of reads unaligned, ∼3.1 million, which explains why it produced the lowest coverage genome. It agreed with the other tools on approximately half of its discarded reads. Minimap2 and NGMLR agreed to leave a further ∼0.5 million reads unaligned, while NGMLR and Winnowmap2 agreed on a separate ∼0.2 million unaligned reads and Winnowmap2 and Minimap2 on ∼0.03 million further unaligned reads.

All tools agreed to leave 682 of the same SMRT reads unaligned. This is because Winnowmap2 excludes less than 1,000 reads in total, where the other tools excluded more reads over roughly three orders of magnitude. NGMLR left the highest number of reads unaligned, ∼200 thousand, which explains why it produced the lowest coverage genome. LRA excluded around 90 thousand reads in all, of which around 70 thousand are common to NGMLR. This also likely contributes to its lower-coverage status shown in Table 3. Minimap2 leaves out just shy of 12 thousand reads in all, most of which are shared with NGMLR and LRA.

In all, not enough information is available at this level of granularity to pass judgment on whether or not tools include or exclude the right reads. For this, an examination of breakpoints is required.

Breakpoints reveal differences between competing alignments of the same genome

Based on the above results, it became clear that we needed to examine the differences in alignment. To compare the alignments to each other, and assess their usability for variant calling across the whole genome, we looked to the SVs known to be present in the NA12878 genome as curated by the DGV resource. We ran a pilot study on now depreciated datasets for linear SMRT data before undertaking a study with current technology (NIST, 2015). The results of these pilot experiments are available as S4 and point to a gap between the largest SV curated in DGV and what is resolvable by long-read sequencing.

To expand upon these initial, promising findings, we accessed the most up-to-date data generated on nanopore and SMRT platforms. Presently, this includes the WGS consortium rel7 on NA12878 for nanopore and a new release of SMRT-CCS on NA24385. NA24385 is not present in DGV, so a high quality callset released by PacBio in collaboration with the Genome in a Bottle Consortium was used as a truthset. These differences are reflected in their labels within Figs. 2 and 3.

Figure 2 Nanopore SV 1,001–10,000 bp.

Variants called by sniffles on each alignment file. (A) Deletions. (B) Insertions. (C) Inversions. (D) Duplications.

Figure 3 SMRT SVs 1001–10,000 bp.

Variants called by sniffles on each alignment file. (A) Deletions. (B) Insertions. (C) Duplications.

We leveraged the sniffles SV caller because of its highly-detailed output files. Our truth sets were the curated SV present in DGV and the set released by PacBio and GIAB for NA24385. Due to the different nomenclature for SV type in the sniffles output and the NA24385 calls, which follow VCF specification (GA4GH, 2023) but differ from DGV annotations, comparisons were limited to the four classes of variant that were most unambiguously labeled in VCFs and the truth sets: insertions, deletions, duplications, and inversions.

The variants available for NA24385 were coded as all insertions or deletions, with subtypes within including duplications. Therefore, the callsets from SMRT do not include inversions as there was no baseline. Sniffles variants were graphed by SV length on the X-axis in shades of blue, grey, and yellow, contrasted with DGV variants in red, organized by platform and SV type. Figures are organized by platform and size of SV, specifically between 1,001 and 100,000 bp, the range at which LRS platforms are known to excel.

Nanopore

For the variants between 1,001 and 10,000 bp (Fig. 2), alignment sets perform largely well on calling deletions that are curated in DGV while inversions were largely missed in all alignments. NGMLR alignments contained the highest number of breakpoints called as duplications. Above the 2001–3000 bp bin, the total duplications in DGV eclipses the number of breakpoints called in any of the alignment files. In contrast, there are very few insertions of this size range present in DGV, but hundreds in our callsets. This could be due to a differential interpretation of the breakpoints, where origin of inserted material has been ascertained (as a duplication) in DGV but remains unattributed (as an insertion) in call sets.

In the 10,001–100,000 bp size range (Supplement S5), DGV largely contains more variants, with the exception of inversions. Inversions present an interesting case here because DGV contains more very large inversions than are reported (>80,001), but various callsets do well at smaller ranges. Across deletions, insertions, and duplications, all alignment tools fall short of the curated reference. It is most interesting that DGV had few insertions smaller than 10,000 bp, but hundreds greater than that threshold. The trend of VCFs from our alignment files overperforming relative to DGV breaks down, and they highly underperform, calling only a few 10s of variants.

SMRT

The truthset for the SMRT dataset from NA24385 does not contain variants annotated as inversions, so Fig. 3 (SV 1001–10,000 bp) and S6 (SV 10,001–100,000 bp) only contain three panels each. We can see immediately in Fig. 3 that the bars generated from the reference set are much closer to the height of the bars from the VCFs, and this is not affected by the use of only the 15 kbp insert set. There is one notable exception with regard to duplications. As with the nanopore dataset, an NGMLR alignment contains the most duplications.

However, these duplications are far in excess of what is present in the reference set. This is notable, as the reference set has been carefully validated. What was missed, versus what is a false positive, is not resolvable in this experiment and likely not resolvable without wet lab bench work or orthogonal validation.

The largest SV show a complex picture (S6). NGMLR calls the most and the largest deletions and no tools do well for large insertions in absolute terms or relative to the truthset. However, Minimap2 and NGMLR preserve no breakpoints called as insertions. For duplications 10,001–20,000 bp, all tools except LRA resolve many more than are present in the reference. NGMLR continues to include many more duplications than the other alignments, and more than the reference.

The disparity between the number of large variants greater 10,001 bp in the NA24385 truthset versus the NA12878 is striking. This, along with the general lack of reference standards for all publicly-available genomes, highlights the need for better, more comprehensive reference sets for all publicly-available resources. Inability to resolve large duplications may yield false negative results for conditions where structural variation is a recognized etiology, such as DMD or FSHD, where variants range between the tens and hundreds of thousands of kbp (Barseghyan et al., 2017; Sharim et al., 2019), beyond the size of variants accurately detected in this study. It is also a critical issue when the technology is used for broad exploratory surveys seeking to identify new etiology in low-complexity regions of the genome where long-read sequencing should shine.

Discussion

In this study we have highlighted the key differences between alignment files generated by different tools. By using well-characterized genome standards, NA12878 and NA24385, we were able to directly compare the performance of each tool on the sequencing datasets obtained on different platforms. We further analyzed the reads that were included or excluded from the alignment, and how these read alignments revealed breakpoints that could be resolved as structural variants in genome architecture. We then compared those variants to sets of previously published variants discovered through multiple platforms.

Reassuringly, each alignment tool was internally consistent: when an alignment tool was given the same fastq and the same reference genome, it produced the same result as judged by bamstats and Sniffles variant callsets. However, when looking across the alignment files produced by different tools on the same sequence data, the representations of the genome diverged in terms of which reads were included or excluded and the numbers and types of variants that were present in VCFs. This is impactful because of the potential high value of LRS data in terms of genome phasing and identification of epigenetic DNA modification (Ni et al., 2019). Since the majority of experiments that leverage large scale population surveys can be expected to rely on reference-guided alignment rather than de novo assembly because of both the cost and speed of analysis (Mardis, 2010), it is key to understand the idiosyncrasies of each type of alignment files prior to generalizing clinical use for these technologies. Furthermore, clinical experiments in genomic medicine face human time constraints—speedier analyses will have higher appeal and adoption.

At this time, GraphMap2 does not show utility for producing whole genome alignments that include structural variations when run with default parameters. The resource usage was large and the time to complete the computation was long and when it did work, Sniffles could not call variants on its product BAM. This is not unexpected, as the tool was designed in large part to increase single nucleotide variant sensitivity in noisy nanopore-sequenced reads.

Minimap2 used the least memory while Winnowmap2 ran successfully in the fastest time, an important point, should these platforms be tied closer to bedside applications. On data from both platforms, they both allowed calling of the most insertions and deletions, but fell short on inversions and duplications.

NGMLR was the most discerning aligner, in that it left the highest number of reads unmapped. It used more compute power than Minimap2 and took much longer (3–10 times) than the next fastest tool. While it was designed specifically to resolve structural variation, it calls a high number of SV that have not been validated with other methodologies or curated in DGV.

There is a great divergence in Sniffles-called variants from alignment files generated by all tools from the variants present in DGV. This is a concerning expansion of seminal findings in a previous study (Chaisson et al., 2019) as none of the Sniffles VCFs mirror the SVs present in the high-quality curated DGV database. Furthermore, there are many genomes released across multiple consortia including GIAB and the Human Pangenome Reference Consortium (Miga & Wang, 2021). But not all of these genomes are sequenced on multiple platforms, and comprehensive reference sets which are harmonious across samples, genomes, and datasets are still limited.

This resulted in us using two reference sets for this project to accommodate the latest releases of data. The callsets generated in our experiments used annotations which were different from the NA12878 variants in DGV and the curated variant set for NA24385, which was surprising. Finally, the most up to date variant benchmark for NA24385 is called on GRCh37-aligned genomes, which is from 2009 and not as comprehensive as current builds (Nurk et al., 2022). Taken together, this underscores the need for orthogonal approaches and collaboration between wet and dry labs to solve this problem.

The points above are critical in designing pipelines for genome analysis and structural variant discovery. In short, it is not a high burden to generate two alignment files per genome with each Minimap2 and Winnowmap2. In computationally unlimited research settings, there is value added in the generation of alignment files from NGMLR as well, although the time to complete these experiments makes them less appealing. These three perspectives on the same genome will account for some of the inherent differences of each tool and the algorithms they use to handle read alignment (Bizjan et al., 2020). If computational resources are limited, Minimap2 is the best choice to move the greatest number of genomes through the pipeline quickly with small memory needs; however, the loss of comprehensiveness must be considered in cases where a suspected variant is not found.

This is impactful in genomic medicine. For example, as variants range from tens of kbp in FSHD and hundreds of kbp to mbp in DMD, diagnosis of these disorders will likely not benefit from data generated on LRS platforms at present, underscoring the need for optical mapping or array-based technologies. However, disorders resulting from smaller SVs such as Huntington’s ( ∼18–540 bp) (Moncke-Buchner, 2002), myotonic dystrophy 1 (∼15–153 bp) and myotonic dystrophy 2 (∼338–143,000 bp) (Yum, Wang & Kalsotra, 2017) could be good candidates for deep study with LRS platforms based on the variants present in alignment files from Minimap2, Winnowmap2 and NGMLR. Accordingly, LRS has been used to identify variants in many such disorders (Mitsuhashi & Matsumoto, 2020).

If pathogenic loci are known, a high diagnostic yield may be obtained by generating maps with each available alignment tool, and use of a structural variant caller such as NanoSV (Cretu Stancu et al., 2017). Unlike sniffles, which provides a call of type of SV (deletion, insertion etc.), NanoSV only identifies breakpoints in the alignment, without assigning those breakpoints an SV type. A robust comparison of SV callers on nanopore datasets highlighted the relative strengths of variant calling pipelines and may help users determine the best caller for their experiments. NanoSV may be suitable for identifying breakpoints missed by Sniffles, but comes with the further caveat that it is resource-intensive and may not scale in a clinical setting without vast computational resources (Zhou, Lin & Xing, 2019).

The discrepancies between the VCFs generated from alignment files starkly show the need to design experiments with the appreciation that the genomics ecosystem cannot yet be dominated by one platform or pipeline and requires a multifaceted approach to discovery. We are at a position where simply because the breakpoint is missing from the VCF from an LRS genome, we cannot say that it is not present. We must therefore look across platforms and data types for comprehensive genome representations (Chaisson et al., 2019). However, this requires more data from the community-generation of NA24385 data on new nanopore chemistry would allow for direct comparison to the current CCS standard, or alternatively new benchmark CCS data could be generated on NA12878. This would address a critical gap in our ability to benchmark new tools in a platform-agnostic manner.

Conclusions

As the cost of long-read sequencing catches up to that of inexpensive short-read sequencing, the inevitable boom in data production will require well thought-out analysis pipelines. Pipeline design always involves a set of tradeoffs. To accurately assess these tradeoffs, we must have a rigorously benchmarked view into the tools available to create the analytic product. Here, we looked at the differences in reference-guided human genome alignments to understand the difference in each tool’s alignment of the same genome, and how it affects a structural variant callset.

This informs our conclusion that, regardless of the sequencing platform, when computational resources are not a limiting factor, it should be best practice to align an LRS human genome with three alignment tools. Minimap2, Winnowmap2, and NGMLR will provide a strong foundation to gain better insight into the architecture of a genome of interest, but there are circumstances where use of LRA for SMRT data may make sense in lieu of NGMLR. When compute resources are limited, minimap2 is a strong choice, and when time is a limiting factor, Winnowmap2 is the best choice.

As we enter the era of multiple references, pangenomics, and graph genomes, analytic substrate from the unmapped reads of a genome gains higher value. Through finding consensus reads unaligned by different aligners, teams interested in data excluded from the reference genome will yield more fertile ground for the discovery of novel genomic material.

Supplemental Information

Supplemental Information 1 LRS Tools

Survey of LRS tools available as of 12 June 2023

Click here for additional data file.

Supplemental Information 2 Coverage statistics

The coverage metrics for each run included in Table 3.

Click here for additional data file.

Supplemental Information 3 Pilot benchmarking

The values for computational benchmarking during pilot experiments on data which have been superseded by new technologies.

Click here for additional data file.

Supplemental Information 4 Benchmarking summary and raw output

The summarized and raw values for computations performed in this article.

Click here for additional data file.

Supplemental Information 5 SV 10,001–100,000 bp

These figures represent the largest structural variants present in callsets from manuscript analyses.

Click here for additional data file.

Adam Kai Leung Wong, PhD, High Performance Computing Specialist for Genomics at the GWU Computational Biology Institute provided critical support in preparing the HPC for this study. The contents of this manuscript are solely the responsibility of the authors and do not necessarily represent the official views of the National Institutes of Health.

Additional Information and Declarations

Competing Interests

Author Contributions

Data Availability

The authors declare there are no competing interests.

Jonathan LoTempio conceived and designed the experiments, performed the experiments, analyzed the data, prepared figures and/or tables, authored or reviewed drafts of the article, and approved the final draft.

Emmanuele Delot conceived and designed the experiments, prepared figures and/or tables, authored or reviewed drafts of the article, and approved the final draft.

Eric Vilain conceived and designed the experiments, authored or reviewed drafts of the article, and approved the final draft.

The following information was supplied regarding data availability:

The code is available at GitHub and Zenodo:

- https://github.com/jlotempiojr/longread_alignment_benchmarking

- jlotempiojr. (2023). jlotempiojr/longread_alignment_benchmarking: Benchmarking long-read genome sequence alignment tools for human genomics applications (1.1). Zenodo. https://doi.org/10.5281/zenodo.8416228

No new sequence data was generated in this study. The data used is available at GitHub:

- https://github.com/nanopore-wgs-consortium/NA12878/blob/master/Genome.md

- https://github.com/human-pangenomics/HG002_Data_Freeze_v1.0

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
