# Peer review of "Benchmarking long-read genome sequence alignment tools for human genomics applications"

_PeerJ, doi:10.7717/peerj.16515_

## Round 0.1 · original submission · Minor Revisions

All three reviewers gave suggestions for modification. Please revise the article carefully.

·

Basic reporting

There are a few places which can benefit from additional citations. These have been noted with line numbers below.

There are few places in the text where the text can be rephrased to reduce ambiguity as well as clarify and structure the content. These have been noted with line numbers in below.

- A rephrase line 34 also might clarify things for the readers - “No alignment from one tool” -> “None of the tools benchmarked independently were able to resolve all large structural variants..”

-Line 73 - there can be an added reference of application of Optical genome mapping to cancer analytics domain

- Some references have the title of the article while some don’t - might need to be fixed e.g. line 104

- Line 139-140 needs a citation for the literature review, so do the following lines where the work is following indications from platform developers. Also a reference to S1 - LRS tools may be needed to define “exclusion criteria” here as it is in the Results section.

-URLs as noted in lines 161-170 are generally not part of research articles, and serve better within the citations. Might be beneficial to check publishing guidelines on citing these effectively.

-Supplemental files are mismatched (Coverage Statistics named file links to the benchmarking stats file and vice versa)

-Samtools (the paper associated) needs a citation on line 233

Experimental design

There are a few ways that the methods can be further described and the supporting evidence enhanced. This includes, but is not limited to - adding some contextual numbers within the text, adding more details around the eventual analytical framework being proposed and some more in-depth investigations to reasons around failures within experiments in certain cases. More details with line numbers are located in below.

-In the abstract (lines 29-32), a rephrase might help structure it better, e.g. “We employed state of the art sequence alignment tools, namely Graphmap2….”, also clarify what it means to have the same “alignment file” each time with a few words.

-Would suggest a better link between the Abstract - Results and Conclusions, and stronger takeaway sentences in the Abstract-conclusion. For instance, “Across the methods benchmark, our results suggest a combined approach by layering alignments from 3 tools will be more effective in generating a complete picture of genomic variability”. The Graphmap2 exclusion may need a brief explanation as well.

-When comparing short-reads, CCS based methods, nanopore etc. a more structured comparison might motivate the gaps better. For instance see the diagram on the pacbio site here (https://www.pacb.com/technology/hifi-sequencing/#:~:text=Explore%20a%20new%20paradigm%20in%20sequencing%20with%20HiFi%20reads&text=PacBio%20sequencing%20technology%20has%20evolved,short%20reads%20and%20Sanger%20sequencing.). A similar figure with the introduction would be helpful in seeing at glance (a) comparison of read length , (b) application and enhanced viability in answering different questions of these technologies. Additionally the mention of Hi-C here seems distracting as the focus of the paper doesn’t address chromatin conformation and this is one of the many different applications of SRS.

-Clarifying statement on why (line 177) the GRCh37 assembly is a possible limitation, is it because of there being corrected and improved versions being available? If so, might be helpful to allude to the changes b/w the latest version and 37 before mentioning that the limitation is minor. (Edit: On reviewing the sections below, it looks like this is because this work used GRCh38 and the ground truth DGV etc. used GRCh37. This is a bit unclear in this section, possibly can be highlighted in line 177).

- Line 279 - Some elaboration on “did not work on our cluster” might be useful here since that seems to be a pretty relevant exclusion criteria, what were the exact reasons it wasn’t able to function on this HPC Environment?

-Line 296/300 - What does “successfully” aligned mean here, is it completion of alignments, or another success metric is used?

-Lines 311 and following will benefit from adding the actual memory utilized. To provide an order of the actual GBs used for each analysis and add context to “little more than 2 times the memory”

-Line 360 and below and Table 3 - This may work better as a heatmap with numbers? While the hues communicate the within row comparison, a more standard way to represent this may be an unclustered heatmap scaled by row. A legend that will be included is also necessary as it is not immediately apparent what the “white” and “light” red columns mean.

Validity of the findings

There are a few papers and links (added below) that the authors may want to peruse and refer to in the manuscript. A valid review of any similar or preceding studies lends credibility to the work here. Additionally I have proposed a few enhancements that will polish and round off the evidence supporting the eventual results. Specific comments with line numbers are provided below.

-It is a bit confusing to list variant calling tools available for LRS and then continue to address the gap in alignment tools. Additionally “Sniffles” is not notated in the citations within lines 97-109. A rephrasing and restructuring of the introduction (towards the end) describing the flow of data in SV calling experiments, a brief annotation of different tooling existing at each step and then specific call outs of differences between alignment tools may be a cohesive motivator for the work. Additionally maybe inclusion of current work here and how this research differs (for instance see paper ref here: https://academic.oup.com/bioinformatics/article/35/17/2907/5298305 which says NGMLR seemed to outperform other aligners – “Depending on the alignment tool used to produce the alignments, SV detection results can vary substantially as Sedlazeck et al. showed for their tool Sniffles (Sedlazeck et al., 2018a). In that study, SV-spanning long reads were aligned with seven different aligners. Their results showed that one particular aligner, NGMLR, outperformed all the others (including BWA-MEM, Minimap2, LAST and BLASR) on the task (Sedlazeck et al., 2018a). In our study, we analyzed read alignments by NGMLR to detect SVs.”

-Line 228 - As all tools were run with default parameters, was there a difference in hardware resource management as dictated by the default params? Threading for instance, do all defaults do single thread operations, maybe possible to explore whether multithreading maintains the ranked order of CPUTime? Possibly something to explore in case it may make a difference.


-Line 289 - Not sure if “reflect typical use in exploratory studies” is a completely accurate assumption, it is not uncommon, and sometimes fairly routine to utilize all capabilities, especially ones that allow for full use of an HPC environment during sequence alignment, or change the output format settings. However, possibly saying “recommended settings” from developers makes a stronger case here.


-Lines 318 - 319 , again to reiterate point above, is there a way to make Minimap2 and others more efficient? Is it possible to explore some tweaking of the command line options to be sure that Winnowmap2 is indeed the most efficient, and it is not just a byproduct of default options. Additionally, linking or referring to the actual command lines/scripts used for each of these (within the github project where it exists) will aid reproducibility.


-Line 326 - How much larger is the nanopore data than SMRT might be a helpful data point here. And any possible parameterizations that can speed things up and allow for a complete result would provide an indication of the kind of changes that might be needed from defaults.


-Line 335 - Did these segfaults result in the middle of the alignment process (possibly from the software encountering something in the data it could not handle), was it a possible resource issue (with Nanopore being much larger) or did it fail at the outset?

-A restructuring and rephrasing of the contents in Line 392 and below (“Breakpoints reveal differences between competing…”) will really enhance the results section. There is a lot of information which does raise a lot of questions when I look at the figures provided (barplots in Fig 2 and 3). Such as - “Why is DGV so low in duplications”, “What does it mean when some of the aligners + variant calling process looks to outperform the truth set?” More restructuring of the text here, which discusses all the relevant plots and their call-outs in organized way, and in addition, editing and polishing the plots that accompany this content (consistency in text of labels (DGV in one graph and truthset in others), removing the axis lines if unnecessary , or putting the legend where it doesn’t overlap the lines, axis labels), with added annotations within the plots or in the plot captions will really strengthen the conclusions derived from the section and make it easier for readers to understand the main takeaways.


-Within the conclusions (e.g. line 529) the authors talk about benefit with different viewpoints produced by different aligners. Is it possible to clarify or sketch an outline of the process the community is recommended to follow when doing so? Is the recommendation a union or intersection of mapped reads? Should researchers be looking at unmapped reads differently depending on how many aligners declared them so? These suggestions and recommendations will really elevate this work to something that can be used fundamentally when selecting the right tools and pipeline for an analysis.


-The conclusion section can also echo an outline of how exactly to utilize multiple aligners in the current LRS analytical landscape. Some specifics here will tie the presented experiments and results together for a reader.

Additional comments

I thank the authors for providing details and experimental results supporting this benchmarking question. I echo their sentiments that this is a very relevant and much-required knowledge gap with the advent of LRS technologies and the known applications. The manuscript was well written and the content addressed the benchmarking questions, as well as presented the evidence needed to add context. To motivate the conclusions and validity further, there are certain possible improvements I would like to suggest (indicated in the sections above as well as some points below). These refer to enhancements that can be made within the text and figures to clarify main takeaways as well as addressing followup questions readers may have about certain parts of the investigation. I have also suggested polishing and adding some of the citations in certain places for completeness as well as some possible extensions of this work that will complement existing results and provide further evidence of some claims. The points specified also contain certain links to other papers, which the authors may want to peruse and refer to in the main manuscript. Adding detail around the framework and specifics of executing bioinformatics pipelines are very important in this research community to help reproducibility and independent verification of results, the points mentioned allude to the changes that can be made in the manuscript to address this need as well.

See below for some additional comments not referenced in the sections above -

-Line 152 - What does “most descriptive” entail, further description might be helpful here.

-What does the “S4” section title on line 181 refer to? Likely this is supplementary data, needs to be clarified?

-Also likely this section can be combined with above as it is mostly linked to data referred to in the “Sequence Data” section.

-Line 224, Possible explanation needed here - Why the use of GraphMapLink, which doesn’t feature in the main set of tools?


-A table within the main manuscript, summarizing the text below might be beneficial (Computational performance and benchmarking).

-The discussion section can be rephrased further to include only highly relevant tools and analysis to the work presented. For instance, using LRS data without alignment (Line 564) is indeed an interesting application, but not relevant to this piece of work.

·

Basic reporting

The paper is well written and addresses an important user need. The authors clearly communicate their goal : they foresee the increased usage of Long Read Sequencing (LRS) and are interested in benchmarking LRS tools for use in clinical analysis pipelines. The primary characteristics of an ideal aligner for large scale human (clinical) genomics are speed and accuracy, which is reflected in the authors' recommendation of aligners to use. The figures & tables are clear (although there could be some minor improvements made as suggested further below), and data & code are provided for reproducibility.

Experimental design

The authors perform a literature review of available aligners and provide a clear rationale for shortlisting 5 state-of-the-art aligners. I appreciate the authors providing details on the version numbers of the tools and the configuration of their cluster, this transparency helps with reproducibility of results. Additionally I commend the authors for running the tools multiple times to prevent bias from a single run, and also running the tools in constrained and unconstrained modes to draw insights from both. Subsequently, the choice of sniffles for SV detection and evaluation makes sense since it is widely used.

Suggested improvements:
"All tools were run with default parameters to reflect typical use in exploratory studies" -- while this is true, since the goal is to benchmark tools to make recommendations for larger studies, I suggest that some amount of parameter tuning could have been used to improve performance of each individual aligner. Ideally, a parameter grid search could have been conducted to find the best parameters to maximize sensitivity and specificity for variant detection. Since that was not provided (and could be a prohibitively large amount of extra work), it would be good to at least include some commentary about the default parameters for each aligner and their effect on each sequencing platform. In general, alignment algorithms should be calibrated to the error profile of the sequencing methodology or it could lead to false positives.

Validity of the findings

The results are clear and the conclusions are well stated. The authors correctly bias towards fast aligners along with those that accurately detect SVs and it is fairly well accepted in the field that minimap2 and winnowmap2 are the state of the art for this task -- so their findings are valid.

Suggested improvements:
It would be good to have a summary table with rows being the aligners and columns being features (memory, speed, efficiency, platform-agnostic, unaligned fraction, etc). This summary table can help with communicating the information at a glance instead of having to scan multiple paragraphs of text.

Apart from the pure counts of SVs detected (as in figures 2 & 3), it would be good to show sensitivity and specificity of the method, ie. what proportion of “true” variants (in each length bucket) were retrieved by each method, and what proportion of variants retrieved by each method are “true” variants.

Additional comments

In the discussion section, please comment on the characteristics of PacBio and Nanopore data that would make it challenging for certain alignment approaches. This would help generalize the learnings from your paper to any new aligners that emerge, or alternatively, provide developers of new alignment tools with ideas on how to fill gaps in the field.

Here are some minor points to be corrected:
l43: Minimap2 misspelling
l73: Please add citation for application of optical mapping in cancer
l353: delete "highlights"?
l395: SVs in plural?
Fig1: Panel A and Panel B seem to be swapped

Reviewer 3 ·

Basic reporting

no comment

Experimental design

no comment

Validity of the findings

no comment

Additional comments

This is very useful study and many researchers will benefit from it. The manuscript is well written and clear. My major complain would be that the benchmarking was done using GRCh38 genome assembly while the complete human genome assemblies by T2T consortium was already available at the time of the study was performed. Below are few minor comments that should help to improve the manuscript.

Minor comments:

1. Sort Table 1 based on tool name (column 1), this will help to find a tool of interest.

2. Since there are many tools for long-reads alignment I’m not surprised that the authors had to limit the study to selected software. However, the selection or rather exclusion criteria are not always clear. For instance, what does exactly mean “Not parameterized for the nuances of current platforms?” This is listed as the reason of not using very popular LAST aligner. In fact, this reviewer is using LAST on daily bases on multiple platforms without any problem. Also, one of the condition of inclusion a software for benchmarking was ability to use the software for both nanopore and PacBio data. However, LRA was used only for PacBio reads. Furthermore, exclusion of the software based on incompatible output (a non SAM format) is rather strange. It shouldn’t be a big deal for a serious bioinformatition to cover an original output to the desired format.

3. Pege 7, line 87. The authors are using “highly homologous pseudogenes” expression. “Homology” means sharing a common ancestry and as such it is a binary function. Two pseudogenes are either homologous or they are not. What the authors refer to should be written “highly similar pseudogenes.” Please correct.

4. There are many unmapped reads, however follow up analysis of these is missing. It would be of great interest to the readers to know what are the reasons for some reads to be excluded from the alignment. For instance, are they low quality reads or contaminants?

---

## Round 0.2 · accepted · Accept

Two reviewers agree to accept. After review, I also think there is no risk of publication.

·

Basic reporting

No comment

Experimental design

No comment

Validity of the findings

No comment

Additional comments

I believe the authors have addressed all my earlier comments with appropriate and detailed changes. I thank the authors for their work on clarifying and elucidating all the relevant points of this work.

Minor point: There may be typesetting fixes required for indentation of paragraphs starting line 179 as they seem to be indented more than previous paragraphs. Some following sections also look more indented than needed. Just confirming the indexing and numbering of sections consistently with the right typesetting would be my suggestion.

Reviewer 3 ·

Basic reporting

No further comments

Experimental design

No further comments

Validity of the findings

No further comments

Additional comments

I'm glad to see the improved manuscript that is acceptable for publication now.